# Enhancing Reliability in ICL: A Mechanistic Analysis of Contrastive Saliency

## Abstract

We explore the impact of different demonstration components on the in-context learning (ICL) performance of large language models (LLMs), focusing on ground-truth labels, input distribution, and complementary explanations. Using explainable NLP (XNLP) methods and saliency maps, we analyze how altering or perturbing these elements affects model behavior. Our findings show that flipping ground-truth labels significantly influences saliency, especially in larger models, while changes to input distribution have a lesser effect. The role of complementary explanations varies by task, offering limited benefits in sentiment analysis but more in symbolic reasoning. These insights are essential for optimizing LLM demonstrations.

## 1 Introduction

Large language models (LLMs) show significant ability of in-context learning (ICL) for many NLP tasks Brown et al. (2020). ICL only requires a few input-label pairs for demonstrations and does not require fine-tuning on the model parameters. However, how each part of the demonstrations used in ICL drives the prediction remains an open research question. Previous works have mixed findings. For examples, although one might assume that ground-truth labels would have a similar impact on ICL as they do on supervised learning, Min et al. (2022) finds that the ground truth input-label correspondence has little impact on the performance of end tasks. However, Zhao et al. (2021) suggests that the example ordering has a strong impact. More recently, Wei et al. (2023) find that only LLMs with larger scales can learn the flipped input-label mapping.

In this work, we use XNLP methods to understand which part of the demonstration contributes to the predictions more. We are interested in the impact of contrastive input-label demonstration pairs built in different ways, i.e., flipping the labels, changing the input, and adding complementary explanations as shown in Fig. 1. We then contrast the saliency maps of these contrastive demonstrations via qualitative and quantitative analysis. Prior works Min et al. (2022); Wei et al. (2023); Brown et al. (2020) show LLMs in relatively small scale, such as all GPT-3 models Brown et al. (2020) (based on categorization in Wei et al. (2023)), cannot override prior knowledge from pretraining with demonstrations presented in-context, which means LLMs do not flip their predictions when the ground-truth labels are flipped in the demonstrations Min et al. (2022). However, Wei et al. (2023) show larger models like InstructGPT (specifically the `text-davinci-002` checkpoint) and PaLM-540B Chowdhery et al. (2022) have the emergent ability to override prior knowledge in the same setting. We partly reproduce the results from previous work Min et al. (2022); Wei et al. (2023) on a sentiment classification task and find that the ground-truth labels in the demonstration are less salient after label flipping.

Meanwhile, as the other important part of the demonstrations, the effect of input distribution is understudied. Min et al. (2022) change the whole input to random words and Wei et al. (2023) do no investigate input distribution at all. Therefore, we investigate the impact of input distribution at a fine-grained level, where we edit the input text's different components in correspondence to task-specific purposes. In the case of sentiment analysis, we change the sentiment-indicative terms in the input text of demonstrations to sentiment-neutral ones. We find that such input perturbation (neutralization) does not have as large impact as changing ground-truth label do. We suspect the models rely on pretrained knowledge to make fairly good predictions because the averaged importance scores for neutralized terms are smaller than the ones of original sentiment-indicative terms. Additionally,

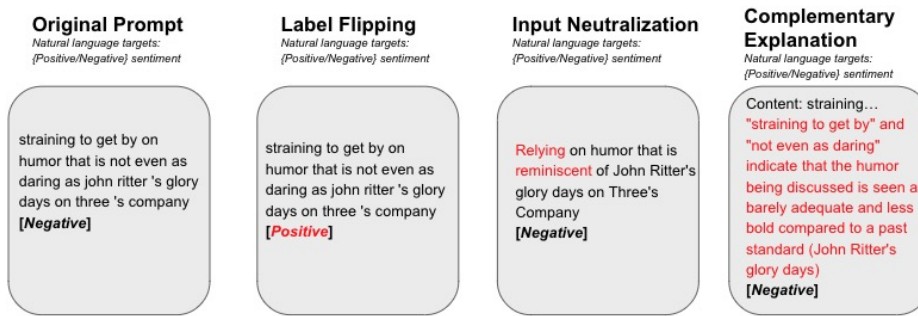

Figure 1: An overview of three ways to build contrastive demonstrations - flipping labels, perturbing (neutralizing) input, and adding complementary explanations. The contrastive parts are colored in red.

we find that complementary explanations do not necessarily benefit sentiment analysis task as they do for symbolic reasoning tasks as shown in Brown et al. (2020), even though the saliency maps suggest the explanations tokens are as salient as the original input tokens. This suggests that we need to carefully generate complementary explanations and evaluate whether the target task would benefit from them when trying to boost ICL performance with such technique.

We hope the findings of this study can help researchers better understand the mechanism of LLMs and provide insights for practitioners when curating the demonstrations. Especially with the recent popularity of ChatGPT, we hope this study can help people from various domains have a better user experience with LLMs. The code for this study will be public once the paper is accepted .

## 2 APPROACH

Previous studies have explored Instruction Consistency Learning (ICL) using traditional methods Min et al. (2022); Wei et al. (2023), but our study is the first to apply XNLP techniques to ICL. We create contrastive demonstrations by flipping labels, neutralizing input adjectives, and adding complementary explanations (see Fig. 1). Our approach differs from Min et al. (2022) in that we employ task-specific input perturbations, focusing on sentiment analysis where adjectives significantly impact predictions. By comparing saliency maps of these contrastive and original demonstrations, we aim to uncover how various demonstration components influence ICL predictions.

## 3 EXPERIMENTAL SET-UP

**Dataset.** We choose **SST-2** Socher et al. (2013), a sentiment analysis task, as our baseline task to explain ICL paradigm. Due to budget limitations and to follow Min et al. (2022); Wei et al. (2023), we randomly sampled 2k examples that are not shorter than 20 tokens from the SST-2 training set as the test set. Additionally, we randomly sample 1k examples for generating saliency maps.

**Demonstration Selection.** We selected four example demonstrations to test language models' in-context learning abilities, including two positive and two negative examples for class balance, as depicted in Fig. 4. These demonstrations involve original texts, label flipping, input neutralization, and adding explanations for each case. **Label Flipping**: We reversed the binary labels for each example for testing. **Input Neutralization**: We tasked GPT-4 to neutralize strong sentiment words in each review, replacing them with neutral alternatives. The changes were minimal and manually verified for accuracy. **Complementary Explanation**: For each demonstration, we generated explanations by prompting GPT-4 to clarify why reviews were labeled positively or negatively, then refined these explanations for brevity and clarity as shown in Fig 4d.

**Baseline LMs and Metric.** We evaluate accuracy of the following models on the sampled SST-2 dataset, including *Fine-tuned BERT, ChatGPT-3.5-turbo, Instruct-GPT, GPT-2*. **Metric:** We use the accuracy to evaluate sentiment classification. We also use T-test to verify our hypothesis on the saliency map patterns for the three contrastive demos.

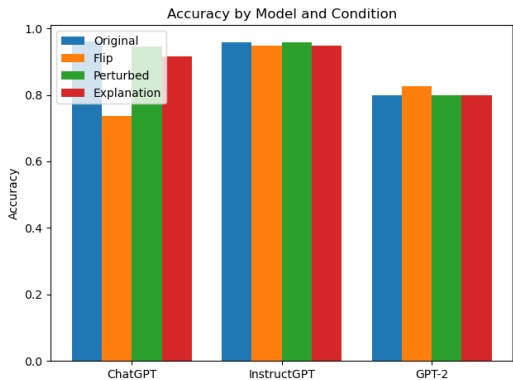

Figure 2: Model Performance under the four conditions, with **four** demonstrations given

**Saliency Map Methods.** TWe utilize Integrated Gradients (IG) Sundararajan et al. (2017) for models like GPT-2, using the Ecco library. For black-box models such as `text-davinci-002` from the Instruct-GPT family, we apply LIME for explanations. We employ LimeTextExplainer, specifying 20 features and 5 neighbors, chosen to minimize API interactions due to budget constraints, resulting in sparser saliency maps discussed in Section 4.2. The hyperparameters and prompts for GPT-2 and GPT-3 are consistent with those used for accuracy evaluation. Due to time and resource limitations, we only produced saliency maps for GPT-2 and GPT-3, with potential future expansions to models like ChatGPT.

# 4 FINDINGS

## 4.1 PREDICTION PERFORMANCE OF THE THREE CONTRASTIVE DEMONSTRATIONS

We evaluated the performance of GPT-3.5-Turbo, InstructGPT, and GPT-2 on test examples with demonstrations like original, label flipping, input neutralization, and complementary explanations, as shown in Fig. 2 and Fig. 3. ChatGPT-Turbo-3.5 showed the most significant performance drop with label flipping, decreasing from 96% to 73% accuracy with 4 demonstrations and further to 17% with 8 demonstrations. InstructGPT experienced smaller drops. Despite similar model sizes, GPT-3.5-Turbo displayed stronger in-context learning compared to InstructGPT.

GPT-2 showed significantly lower performance with 4 demonstrations and tended towards negative predictions with 8 demonstrations, indicating insensitivity to demonstration type contrasts. This supports previous findings that large LMs like ChatGPT and InstructGPT are more affected by label flipping in demonstrations.

Input neutralization and complementary explanations had minor impacts on model performance, likely due to the trivial nature of the sentiment analysis task and the models' reliance on pre-trained knowledge. This leads us to further explore contrasting saliency map patterns between smaller and larger LLMs, all based on transformer architecture."

## 4.2 COMPARISON OF THE SALIENCY MAPS

Due to the GPT-2's poor performance and compute cost when given 8 demonstrations, we use the setting of 4 demos for saliency map in Fig. 4 and Fig. 5.

**Label Flipping.** The labels in the demonstration are less important after model flipping for smaller LMs (GPT2) but more important for large LMs (`text-davinci-002 from Instruct-GPT`). For example as in Fig. 4a and Fig. 4b, the importance of the output label in the demonstration decreases from the original prompt to the label-flipped one. This suggests that the model might pay less attention to the flipped label due to its inconsistency with the input, which results in insensitivity to label flipping in the demonstrations. We expect smaller LMs (GPT2) and large LMs (`text-davinci-002` from Instruct-GPT) to have different behaviors because Wei et al. (2023) show only large LMs have the ability to override prior knowledge from pertaining to the one from demonstrations, which is also supported by our results from Fig. 2 and Fig. 3.

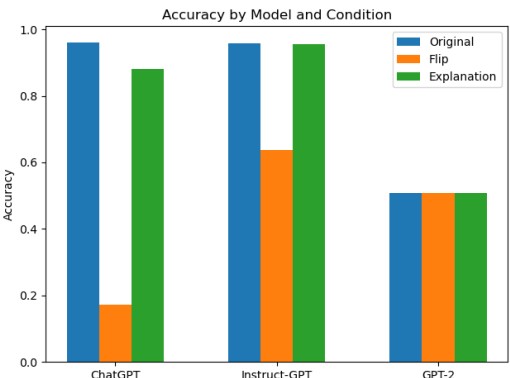

Figure 3: Model Performance under the four conditions, with **eight** demonstrations given.

For GPT2, on average, $3.35/4$ of the labels in the demonstration have decreased saliency scores when the demo labels are flipped. Moreover, the average saliency scores of the 4 demo labels **decrease** for all 20 test examples. The p-value from a T-test for comparing average saliency scores ($N = 20$) between original and label-flipped demonstrations is $< 0.001$. For InstructGPT, the average saliency scores **increase** for $16/20$ test examples with a p-value of 0.23 from a similar T-test as above (Fig. 5b). As InstructGPT achieves around $60\%$ accuracy in Fig. 3, we expect Instruct-GPT (with 8 demonstrations) and ChatGPT to have a more significant result as it shows the ability to fully override prior pretrained knowledge.

**Input Perturbation (Neutralization).** The sentiment-indicative terms in the original prompt are more important than sentiment-neutral terms in the neutralized prompt. The hypothesis is derived from the definition and our intuition of the sentiment analysis task. Sentiment-indicative terms are important to make sentiment predictions. To validate this hypothesis, we contrast the original and neutralized prompts and manually pick different tokens with sentiment orientations. The selected tokens are highlighted in Fig. 4a and Fig. 4c with red boxes respectively. We then compute the average saliency scores for each of the 20 test examples.

We find that, for GPT2, the average saliency scores for sentiment-indicative terms in the original prompt are higher than their contrastive parts in the neutralized prompt for all 20 test examples with a p-value of $< 0.001$ from a T-test. However, for Instruct-GPT, we find that the sentiment-indicative terms in the original prompt are equal or higher in $9/20$ test examples with a p-value of 0.17 from a similar T-test as above. We note that, as mentioned in Section 3, the saliency maps for Instruct-GPT generated by LIME are sparse and have a lot of zeros as shown in Fig. 5. This may lead to a mixed result with a less significant T-test result.

### 4.2.1 COMPLEMENTARY EXPLANATION

Previous research Min et al. (2022) demonstrates that complementary explanations aid symbolic reasoning tasks like Letter Concatenation, Coin Flips, and Grade School Math. However, our findings in Fig. 2 reveal that these explanations do not enhance sentiment analysis, a relatively simpler task for language models. Saliency maps for GPT2 indicate that, in 80% of cases, explanation tokens have higher saliency scores than review tokens, with review scores averaging 90% of explanation scores, underscoring their comparable importance. The effectiveness of complementary explanations appears task-dependent, benefiting tasks that require logical reasoning. Further research is needed to confirm this across more datasets, which we suggest for future studies.

## 5 CONCLUSION

In this study, we applied XNLP techniques to explore ICL by analyzing contrastive input-label pairs with added explanations and examining their saliency maps through qualitative and quantitative methods. We partially replicated prior findings on a sentiment classification task, noting that ground-truth labels become less salient after label flipping. Neutralizing sentiment-indicative terms in inputs impacts model performance less than label changes, suggesting reliance on pretrained knowledge, as shown by lower importance scores for neutralized versus original terms. These insights aim to enhance understanding of LLM mechanisms and guide practitioners in demonstration curation.

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

# A APPENDIX

## A.1 RELATED WORK

Large language models (LLMs) show significant ability of in-context learning (ICL) for many NLP tasks. Min et al. (2022) show that presenting random ground truth labels in the demonstrations does not substantially affect performance. They also change other parts of the demonstrations (e.g., label space, distribution of the input text and overall sequence format) and find these factors are the key drivers for the end task performance. Wei et al. (2023) concentrates on labels by comparing LMs across different size scales with two variants that have flipped labels or semantically-unrelated labels. They find that only large LMs can flip the predictions to follow flipped demonstrations. Akyürek et al. (2022) try to understand in-context learning by training transformer-based in-context learners on small-scale synthetic datasets.

### A.1.1 GRADIENT-BASED METHODS

For models with parameter access, we can estimate the importance of an input token using derivative of output w.r.t that token. The most basic method assigns importance by the gradient. However, it suffers from some known issues such as sensitivity to slight perturbations, saturated outputs, and discontinuous gradient. SmoothGrad Smilkov et al. (2017) reduces the noise in the importance scores by adding Gaussian noise to the original input. Integrated Gradients (IG) Sundararajan et al. (2017) computes a line integral of the vanilla saliency from a baseline point to the input in the feature space.

### A.1.2 PERTURBATION-BASED METHODS

An alternative approach to generating saliency maps using input perturbations can be applied to black-box models. Instead, the process involves systematically altering the input data (i.e., words, phrases, and sentences) and observing the changes in the model's output. We plan to start with the standard method that falls into this category, LIME Ribeiro et al. (2016). The process involves creating perturbed versions of an input instance, passing them through the model, training a local linear model on the perturbed inputs and their corresponding predictions, and extracting feature importances from the local model.

(a) Original prompt

(b) Prompt with label flipping in the demonstrations

(c) Prompt with input perturbation (neutralization) in the demonstrations

(d) Prompt with complementary explanations in the demonstrations

Figure 4: Full prompts (demonstration + test example) used for original demonstration and three contrastive variants. Tokens are color-coded by saliency scores for GPT2 generated by IG. The red box in original and neutralized prompts indicates manually selected sentiment-indicative and sentiment-neutral terms that we used for saliency map comparison.

Review: straining to get by on humor that is not even as daring as john ritter 's glory days on three 's company .
label: negative
Review: , serves as a paper skeleton for some very good acting , dialogue , comedy , direction and especially charm .
label: positive
Review: a whole lot of fun and funny in the middle , though somewhat less hard-hitting at the start and finish .
label: positive
Review: might have been saved if the director , tom dey , had spliced together bits and pieces of midnight run and 48 hours ( and , for that matter , shrek )
label: negative
Review: a movie that successfully crushes a best selling novel into a timeframe that mandates that you avoid the godzilla sized soda .
label: negative

(a) Original prompts (demonstration + test example) used for original demonstration (Instruct-GPT)

Review: straining to get by on humor that is not even as daring as john ritter 's glory days on three 's company .
label: positive
Review: , serves as a paper skeleton for some very good acting , dialogue , comedy , direction and especially charm .
label: negative
Review: a whole lot of fun and funny in the middle , though somewhat less hard-hitting at the start and finish .
label: negative
Review: might have been saved if the director , tom dey , had spliced together bits and pieces of midnight run and 48 hours ( and , for that matter , shrek )
label: positive
Review: a movie that successfully crushes a best selling novel into a timeframe that mandates that you avoid the godzilla sized soda .
label: negative

(b) Prompt with label flipping in the demonstration (Instruct-GPT)

Review: Replying on humor that is reminiscent of john ritter 's glory days on three 's company .
label: negative
Review: Serves as a paper framework for some standard acting, dialogue, comedy, and charm.
label: positive
Review: Generally average and neural in the middle, albeit slightly less impactful at the start and finish.
label: positive
Review: The movie may have been different if the director, Tom Dey, had incorporated elements of Midnight Run, 48 Hours (and, for that matter, Shrek).
label: negative
Review: a movie that successfully crushes a best selling novel into a timeframe that mandates that you avoid the godzilla sized soda .
label: positive

(c) Prompt with input perturbation (neutralization) (Instruct-GPT)

Review: straining to get by on humor that is not even as daring as john ritter 's glory days on three 's company .
Explanation: "straining to get by" and "not even as daring" indicate that the humor being discussed is seen as barely adequate and less bold compared to a past standard (John Ritter's glory days)
label: negative
Review: , serves as a paper skeleton for some very good acting , dialogue , comedy , direction and especially charm .
Explanation: "very good acting", "dialogue", "comedy", "direction", and "especially charm" are generally associated with positive sentiments in the context of a review.
label: positive
Review: the work of a filmmaker who has secrets buried at the heart of his story and knows how to take time revealing them .
Explanation: it praises the filmmaker's skill in creating intrigue and suspense, suggesting a well-crafted and engaging story. "has secrets buried" and "knows how to take time revealing them" indicate a mastery of storytelling, which is generally viewed as a positive quality in filmmaking.
label: positive
Review: might have been saved if the director , tom dey , had spliced together bits and pieces of midnight run and 48 hours ( and , for that matter , shrek )
explanation: it implies that the director's work was unsatisfactory and the film could have been better if it had incorporated elements from other successful films, suggesting that the film as it stands is not good enough.
label: negative
Review: a movie that successfully crushes a best selling novel into a timeframe that mandates that you avoid the godzilla sized soda .
label: positive

(d) Prompt with complementary explanations in the demonstrations (Instruct-GPT)

Figure 5: Full prompts (demonstration + test example) used for original demonstration and three contrastive variants. Tokens are color-coded by saliency scores for generated by LIME. The red box in original and neutralized prompts indicates manually selected sentiment-indicative and sentiment-neutral terms for saliency map comparison.

