# OpenReview forum: "Enhancing Reliability in ICL: A Mechanistic Analysis of Contrastive Saliency"
_ICLR.cc/2026/Workshop/Sci4DL — Submitted to Sci4DL 2026_

### Official Review · Reviewer_eczs · 2026-02-27

**Fit:** 1
**Significance:** 1
**Confidence:** 2

**Summary:**

This paper appears to be almost an exact match to the 2023 paper titled “Towards Understanding In-Context Learning with Contrastive Demonstrations and Saliency Maps,” which I remember reading and using before. The core questions are still important. Understanding which components of in-context learning actually influence model behavior is still a relevant and meaningful research direction. However, the methods used here feel dated and weaker by current standards.

The paper studies three types of manipulations to in-context demonstrations. First, flipping ground-truth labels. Second, perturbing the input distribution by neutralizing sentiment-indicative words. Third, adding complementary explanations. The authors then analyze model behavior primarily through saliency maps, using Integrated Gradients for GPT-2 and LIME for InstructGPT. They treat changes in saliency as evidence for how different components of demonstrations influence predictions.

The main claim is that flipping labels significantly changes saliency patterns, especially for larger models, while input neutralization has less effect. Complementary explanations appear to provide limited benefit for sentiment analysis. The experiments are conducted on SST-2 and use relatively early-generation models such as GPT-2 and InstructGPT.

While the framing is clear, I feel that the study does not hold up given the current state of the field, both methodologically and conceptually.

**Strengths:**

One strength of the paper is that the question it asks is still valid. It is reasonable to want to decompose in-context learning into components such as labels, input distribution, and explanations. That framing is useful and aligns with broader discussions in the literature about label space, formatting, and task adaptation.

Another strength is that the manipulations are concrete. Flipping labels, neutralizing adjectives, and adding explanations are easy to understand and easy to replicate. The setup is transparent, and the authors are clear about what they are changing and how they are measuring effects.

I also appreciate that the paper attempts to go beyond pure accuracy and instead uses saliency to inspect what the model is attending to. Even if I disagree with saliency as the primary method, the intention to analyze internal behavior rather than only surface metrics is good.

**Suggestions:**

Weaknesses

My main issue is conceptual and methodological. I am not convinced (or believe) that label flipping is a meaningful contrastive demonstration. Flipping labels ends up creating incorrect demonstrations. It does not necessarily isolate a component of in-context learning. It may instead introduce a direct clash with the model’s pretrained knowledge. If saliency changes under flipping (which it seems to in the paper), it would be unclear whether that reflects in-context adaptation or a knowledge conflict. The paper does not clearly distinguish between those two explanations (I am not claiming that these are the only 2 reasons).

The second major weakness is methodological. Saliency maps (in my opinion and to my knowledge) are no longer considered strong mechanistic evidence. They are correlational and sensitive to implementation details. There are now more interventional and targeted approaches such as activation patching, representation geometry analysis, controlled synthetic tasks, or even logit lens analysis that produce slightly more causal claims.

The models used are also outdated by current standards. GPT-2 and InstructGPT are no longer representative of modern LLM behavior. Given that the field has evolved significantly, the model choices feels behind the state of the art; and there is no way to know if the current models demonstrate similar behaviors.

Another issue is the task choice. SST-2 is highly saturated. LLMs have almost certainly seen vast amounts of sentiment data during pretraining. It is unclear whether the demonstrations are doing meaningful work or whether the model is relying on prior knowledge. This makes it difficult to interpret any findings as specifically about in-context learning rather than pretrained competence.

⸻

Suggestions

I think the core idea could still be developed into something stronger.

First, the paper should reconsider the use of label flipping as the primary contrastive condition. Instead of simply introducing incorrect demonstrations, it would be more meaningful to design contrastive settings that isolate specific mechanisms as to why certain components of ICL are being used, such as introducing genuinely new information, correcting outdated priors from pretraining, or requiring new composition of knowledge.

Second, the methodological approach should move beyond saliency.

Third, the task design should vary in terms of saturation and difficulty. It would be much more informative to compare highly saturated tasks with less common or compositional tasks, and to analyze how different model sizes/different model types respond to those variations. The number of demonstrations and the density of information in the prompt could also be systematically varied rather than just the contrastiveness of the components.

---

### Official Review · Reviewer_yfxi · 2026-02-28

**Fit:** 2
**Significance:** 1
**Confidence:** 2

**Summary:**

The paper studies the behaviour of in context learning in response to simple perturbations to the demonstrations for a sentiment analysis task. On this task, the experiments reveal that flipping the label leads to significant response with response varying across larger and small models.

**Strengths:**

1. The motivation of the work is sound; it certainly is scientifically interesting what mechanisms drive in-context learning.

**Suggestions:**

However, the paper is significantly lacking in the current form to conclusively claim interesting insight. In contrast to Min et al, Wei et al. (also cited in the paper), the only novelty I see is the adoption of explanation techniques. Plus the task studied is simple, and the perturbations are also simple. While simple studies cam definitely be scientifically interesting, my reading of the paper does not reveal interesting insight that is not already covered by the prior literature. As a recommendation, I'd advise the authors to further strengthen the scope of the study (and the experiments).

---

### Meta-Review · Area_Chair_dw7c · 2026-03-02

**Recommendation:** Reject

**Metareview:**

The work studies the behaviour of In-Context Learning in a sentiment analysis task when applying simple perturbations to the data/context. Although the research question is important and interesting, both reviewers agree that similar analysis already appeared in the literature, and that the paper in its current form does not add additional insights into its conclusions or methodology.

---

### Decision · Program_Chairs · 2026-03-02

Reject